# Integrating Prescriptions into Calendars: Design Considerations

Maybins Lengwe*
University of Victoria

Jens Weber†
University of Victoria

Charles Perin‡
University of Victoria

## ABSTRACT

Optimizing the adherence to medical prescriptions remains an ongoing challenge in our health care systems. Among the many reasons for non-adherence (including socioeconomic status) are forgetfulness and the need to manage dynamically evolving busy lives. Electronic medication reminder apps have become broadly available to aid this situation. However, these apps are rarely integrated with other tools used for managing day to day activities, e.g., general electronic calendars. In this paper, we investigate the design of calendars that can also be used for managing medical prescriptions. The proposed designs include mechanisms for visualizing medication entries and communicating attributes related to medication prescriptions. The designs also include ways of annotating conflicts arising from scheduling of medication entries that violate temporal constraints specified in the prescriptions. A user study was conducted to evaluate the calendar designs. The results indicate potential benefits and considerations for utilizing calendars to manage prescriptions.

**Index Terms:** Visualization—Visualization design and evaluation methods—Calendar Design—Prescription Schedules.

## 1 INTRODUCTION

A prescription is a common and important form of medical intervention provided a clinician to a patient [1]. It indicates actions such as taking medications, following a diet, or executing physical exercises [2]. When agreed upon between a patient and their healthcare provider, the patient is expected to follow their prescription [3]. The extent to which a patient follows an agreed-upon prescription is referred to as *adherence* [2]. Non-adherence to prescriptions is a significant problem in healthcare [2, 4]. Adherence rates average 50% and account for 33-69% of hospital re-admissions, resulting into billions of dollars per year [5, 6].

Non-adherence is a multidimensional phenomenon, influenced by a complex interplay between socioeconomic, healthcare system-related, medical condition-related, therapy-related and patient-related factors [7]. Patient-related factors of non-adherence include fear of side effects, cost implications, too many medications, failure to perceive benefit, and mistrust of prescribing physicians [8–10]. Non-adherence can also result from forgetfulness and inadvertent omissions that result from failure to make lifestyle changes that take the prescriptions into consideration [11, 12]. Forgetfulness is often addressed through reminders that trigger alarms when a patient is supposed to take medications [6, 10, 13–15]. But for patients with frequent and dynamically changing schedules, static reminders often fall short of addressing the problem. Consider a patient who manages a busy schedule of dynamically changing day-to-day commitments using an application (typically a calendar), while using a separate medication reminder app. What happens when a medication reminder is triggered during the time when the patient is busy? How does that patient reschedule the reminder taking into considerations

*e-mail: mlengwe@uvic.ca
†e-mail: jens@uvic.ca
‡e-mail: cperin@uvic.ca

other activities already planned for? How does one resolve clashes between other activities and medication administration without violating the constraints specified in the prescription? An integrated redesign of these apps is necessary to address these questions.

Electronic calendars have become instrumental in the management of daily activities [16–18]. They are used to coordinate interactions among individual schedules of family or team members and can convey meaning and values behind the priorities of scheduling [19]. Calendars have been used to visualize temporal trends that include everyday activities such as energy use in work places, fitness tracking, and work routines [20–22].

We are interested in exploring the possibility of integrating prescription management into electronic calendars that are already used by many patients [23, 24]. Such integration comes with challenges. The first challenge is how to render the prescription entries so that the user can differentiate between a normal calendar entry and one that is part of a prescription. The second challenge is how to ensure that patients who reschedule prescriptions do so within the specified safety constraints.

Our design goal is to integrate prescriptions into calendars with support for safety checks that avoid (or at least indicate) changes to schedules that no longer adhere to prescribed constraints (**DG1**). We consider ways of rendering medication entries so that they are easily identifiable and that all medication-related information is clearly conveyed. We explore ways of rendering unsafe drug-interactions that may arise when medication administration times are being scheduled or changed. This is an important aspect to consider because administration times that conflict with a prescription are considered a form of non-adherence [25]. To our knowledge, incorporating prescriptions and reminders into calendars has not been studied before.

Our contribution is twofold. First, we identify and discuss considerations for the design of calendars that support medication prescriptions. Such calendars allow for the scheduling of medication actions alongside other everyday activities and provide a way for rendering and resolving conflicts when they are raised by unsafe schedules (i.e., schedules that violate constraints specified in the prescriptions). Second, we present the results of a qualitative study with twelve participants interacting with alternative calendar designs. Results indicate the potential benefit of equipping electronic calendars that already in use by many patients with additional functionality to support the scheduling of medication prescriptions. Users are generally in favour of using such an integrated approach that leverages their familiarity with existing tools. Results also show that it is feasible to design calendars that effectively communicate unsafe medication schedules. These results inform five additional design goals that an integrated calendar should address: the use of familiar design (DG2), avoiding clutter (DG3), allowing for personalization (DG3), supporting personal reflection (DG5), and highlighting for user attention (DG6).

In the remainder of this paper, we first provide a review of relevant literature. We then describe the data format of medication prescriptions and basic usability requirements for an on-calendar prescription visualization, before introducing three alternative designs. We subsequently report on the design of our user study, its methodology and our results, before deriving a set of design considerations for integrating prescriptions into calendars.

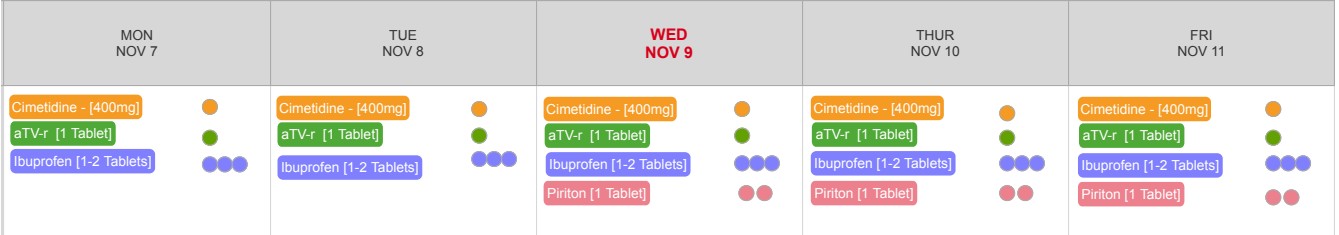

Figure 1: Calendar design with daily medication summaries that giver users a preview (Name and number of times) of the medication to be administered on a particular day.

## 2  RELATED WORK

We review the literature relevant to on-calendar prescription visualization. This includes work related to medication reminders, visualization of schedules, and on-calendar visualizations.

### 2.1  Medication Reminders

Reminders are among the most common technological interventions to improve adherence to medication [4, 26, 27]. Reminders can take many forms, including interventions of caregivers through video and voice calls [28] and text messages [29], smart pill boxes [10], and computer applications [13]. Focusing on systems that visually represent reminder events to stay within the scope of this research, we identified two relevant technological interventions.

The first one is a health literacy tool called Medication Calendar [14]. It was designed to improve antihypertensive medication adherence. The Medication Calendar provides a graphical view of medication to be taken during a given section of a day. Its layout shows Morning, Afternoon, Evening, and Bedtime as columns and the medications as rows. It displays (in text format) the name of the medication, the time of day that it should be administered, the number of times daily that it should be taken, dosage information, and additional clinical indications. The application then triggers reminders when it is time to administer the medication.

The second relevant intervention is a tool for representing graphically enhanced interventions for coronary heart disease [30] The tool shows the time of day (Morning, Afternoon, Evening, and Bedtime) as columns and the list of medication as rows. An additional column indicates the purpose of the medication. Row headers include medication name, dosage, and the time of day when it should be administered. The novelty in this work is that the table cells contain graphical images of the corresponding medication.

These two contributions provide foundational design elements to consider for visualizing medications and their reminders in a calendar-style design. Specifically, the layout (columns and rows) and design concepts (graphical representation of drug entities) provide a starting point for an integrated design.

### 2.2  Visualization of schedules

To introduce medication reminders and schedules in general calendar applications, we turn to related work on visualization of schedules. Defined broadly, visualization of schedules involves the representation of planned events on a timeline that depicts a sequence of events (where each event has a single time point) or interval event (continuous quantitative time-series data) [31]. The events are often represented as bars that span multiple time points or nodes that are attached to single time points.

Gantt charts [32] and Pert charts [33] are two standard ways of representing event schedules that have been extended (addressing issues such as visual clutter and event prioritization) to solve planning problems in the field of Engineering (e.g., [34–36]). While not exclusively focusing on visualizing *planned* events, in the healthcare context a large number of systems have been developed for visual analysis of patient's cohorts temporal data (that also consist of series of temporal events). Lifeline [37, 38], Lifeflow [39], Prima [40] and TimeSpan [41] are some of the works that have explored visualization of timelines to show patients' temporal information such as medication histories, hospital visitations, and treatment processes. Walker et al. [42], developed patients' visualizations for emergency department wait times.

In this research, we focus on i) visualizing a single patient's data, and ii) visualizing events on typical calendar layouts, which often consist of two-dimensional charts where one dimension shows days and the other dimension shows the time of day [43]. Alternative layouts exist and we discuss them in the next subsection.

### 2.3  On-Calendar Visualization

Two dominant models are used to visualize events on a calendar: the radial model and the linear model [44]. With the radial model, data points are positioned along a circle, ellipse, or spiral [45]. Lines are usually drawn from the center of the circle to its circumference at equal adjacent spacing to represent time points. Rings drawn from the center of the circle extending outwards are added to further divide the temporal dimension. Popularized by William Playfair [46] and Florence Nightingale [47], radial calendars have proven effective in visualizing univariate calendar events. Variations of these include Radar Bars [48] and Radar Plots [49]. Radial calendars have been used to show daily consumption of provisions of everyday supplies for a whole calendar year by varying the color and size of marks along the rings [44]; to visualize personal data obtained from different body sensors [50] by positioning stacked bars on a single ring with 24 partitions for a 24-hour day; or quantified-self data [51] by showing streamgraphs and heatmaps of multi-year data along a spiral. Although hailed as state-of-the-art [48] and space-efficient [52], the radial model has limits when it comes to representing multivariate data and data without a defined time limit.

The linear model does not suffer from the same limitations as the radial model. It entails the conventional calendar layout with rows and columns representing days and times of day (or vice versa), and each cell representing a time interval of a particular day [53, 54]. Huang et al. incorporated quantitative user feedback data, collected from daily activities using Fitbit, within a personal digital calendar [21]. Basing their design on Google calendar [54], they overlaid the calendar with horizontal line graphs to show a user's level of activity each day. Wijk et al. used a linear calendar to show both the power demand and employee presence at a research facility [20] using color hue, height, and color saturation. MineTime Insight [55] is a tool for improving short and long-term scheduling decisions. It uses a calendar design to show multiple coordinated views for the exploration of personal calendar data. This tool however, concentrates on the analysis of calendar data and not on augmenting the calendar with additional information, such as medication events. Researchers have also investigated shared calendars [56], and more recently, how calendars can be used to enhance team communication and collaboration [57]. In the latter, they augmented a calendar

with visualization support for the exploration of conversation data generated by team messaging platforms like Slack. They used color to identify keywords in messages, size to indicate word frequency, and lines to show connections between messages within the channel.

Incorporating non-standard data, such as fitness data or daily activities, on a calendar view has been studied at length [44, 50, 55–57]. However, these efforts do not fully support the visualization of medication schedules requirements because most research efforts i) focus on representing univariate and quantitative data; ii) tend to ignore the standard calendar events already present; and iii) are usually limited to supporting events with only two attributes (time and a quantitative attribute) [23, 37, 41]. Visualizing events with a temporal and a quantitative attribute is usually achieved through adding line graphs [21], shading [57], or bars [50] to the calendar. Those are the only visual elements added to the calendar view. Our work differs from this scenario because in our context, events have more than two attributes (allowed placement interval, preferred time, duration, and other descriptive units), are to be presented alongside the normal calendar events, and mechanisms are needed to visually communicate conflicts between events. To the best of our knowledge, there is no existing calendar design that presents a way of visually communicating conflicts that may arise between two or more events, and visualizing prescriptions with their temporal constraints alongside other activities in a standard calendar has not previously been investigated.

## 3 MEDICATION PRESCRIPTIONS

To design an on-calendar visualization of prescriptions, we need to i) understand the relevant characteristics of a prescription; and ii) the usability requirements such a system should support.

### 3.1 Prescription Characteristics

According to Ethier et al. [58], a prescription is made of the different parts which define the modalities of administration for a given drug. The underlying building block is the drug prescription item. The drug prescription item comprises the drug administration specification, healthcare objective specification, and drug distribution specification. The drug administration specification is the part that contains information that pertains to drug administration. This information includes the drug product specification (which indicates the name of the drug), drug dosage specification (which indicates the dosage, administration route, and dosing conditions), and drug course specification (which indicates the the starting condition and the duration of the prescription). Kumar et al. [59] summarizes these building blocks as superscription (directive to take), inscription (name and dose), subscription (directions to the pharmacists), and signature (instructions for Patient), and Fox [60] as drug name, drug dose, drug dose units, drug dose frequency, duration (comprising start and end date), and indication. Like others (e.g. [14, 30]), we adopt the latter classification by Fox [60] because of the simplicity and directness of the naming convention.

Following this classification, the design of an on-calendar prescription visualization should support medication entries with the following attributes: Drug name (D1), Drug dose (D2), Drug dose frequency (D3), Duration (D4), and Drug indications (D5). These attributes are all employed by prescriptions in both short term (acute) and long term (chronic) disease management [61].

As an example of how these data items exist in a prescription, let us consider the following scenario presented by Diemert et al [62].

> *Ms. Smith's physician has prescribed 6mg of Warfarin to be taken orally once daily to address her deep vein thrombosis. Additionally, she was prescribed 15 mg of long-acting Morphine orally twice daily and 600 mg of Ibuprofen to be taken three times a day as needed with food to help manage her pain.*

The scenario above specifies three different medications: Warfarin, Morphine, and Ibuprofen. The classification above describes the prescribed Ibuprofen, for example, with a name (Ibuprofen - D1), dosage (6mg - D2), frequency (three times a day - D3), duration (no end specified - D4), and indication (with food - D5). In the rest of the document, we use the term *medication* more often than the term *drug* because it refers more broadly to an entry in a prescription [62]; it also includes prescription of physical activity, for example.

### 3.2 Usability Requirements

An on-calendar prescription visualization should support tasks that relate to reading calendar entries (e.g., reading calendar events and accessing event information) and to managing calendar entries (e.g., adding, editing and deleting calendar entries).

Representing medication prescriptions in general-purpose calendars requires adding visual information about these prescriptions on top of an existing planned schedule. This means that such a calendar must support tasks related to reading calendar entries (basic function of a calendar) as well as reading information about prescriptions and their potential conflicts.

Tasks related to reading standard calendar entries have been discussed in previous work [63–65]. These include tasks related to retrieving temporal features and reading event-related information such as date, time, location and purpose. From this, we derive the following *basic* usability requirement:

> $R_{layout}$: *The user should be able to correctly and efficiently read the calendar's temporal features. These features include the calendar's current day, month and year; the days of the week; and the times of the day.*

The addition of medication entries in the calendar introduces new visual elements. Medication entries communicate more details (D1 - D5) than the standard title, time, and location of regular calendar entries [66–68]. Integrating prescriptions in a general-purpose calendar involves integrating Personal Health Information (PHI) and related activities into calendars [69–71]. Therefore, a design that integrates prescription visualization to a calendar should satisfy the following *prescription-related* usability requirement:

> $R_{medic}$: *The user should be able to accurately and efficiently identify the calendar's medication entries. They should also be able to read the entries' name (D1), dosage (D2), frequency (D3), duration (D4), and indications (D5).*

Prescriptions come with constraints, e.g., drug dosage, administration frequency and other indications. Constraints may relate to a single medication or a set of different medications. An example of within-medication constraints is found in the the prescription *Take 600 mg of Ibuprofen three times a day as needed with food*, that has three constraints: 1) that 600mg should be taken at a given time; 2) that the maximum number of intakes per day is three; and 3) that the drug must be taken with food. But medications are often more complex. Consider for example the following prescription: *Take Tenofovir 1 tablet once daily and Metformin 500mg once daily. Take Tenofovir and Metformin at least 6 hours apart*. This example includes within-medication constraints: both Tenofovir and Metformin must be be taken once daily, and the dosage is 1 tablet for the former and 500mg for the latter. This example also includes a between-medication constraint: the two medications should be taken at least 6 hours apart. A violation of constraints (within- or between-medications) is what we call a scheduling conflict. In this paper, we focus on conflicts that deal with restrictions in the scheduling times of medications that may be either unsafe or ineffective when taken together. The nature of these conflicts falls into two categories: too close (those with lower limit time constraints) and too far apart (those with upper limit time constraints). Previous work

Figure 2: $Design_A$ with colored vertical bars used for medication entries and gray rectangular entries used for other activities.

has highlighted that a calendar that supports event attributes should also have mechanisms for dealing with conflicts that arise between entries [72]. This is true for medication entries. Therefore, a design that integrates prescription visualization to a calendar should satisfy the following *conflict-related* usability requirement:

> $R_{conflict}$: *The users should be able to correctly identify conflicting medication entries in the calendar. They should also be able to name the medications involved in the conflict, identify the nature of the conflict, and envision actions that may be taken to resolve the conflict.*

These three usability requirements are basic requirements that an on-calendar prescription visualization should satisfy. Next, we present three calendar designs that were created to satisfy these requirements.

## 4  On-Calendar Prescription Visualization Design

In this section, we propose three designs that show the data (D1 - D5) and were created to meet the usability requirements $R_{layout}$, $R_{medic}$ and $R_{conflict}$.

### 4.1  Ideation Session

We started the design process with an ideation session involving eight researchers with background in human-computer interaction and visualization. The goal of the session was to explore layout variations and features that a calendar with integrated medication prescriptions should have. We asked the researchers to sketch calendars that show all the data (D1 - D5) and satisfy the given requirements. One ideation session was held for this task and it lasted for 30 minutes. Participants used pens, colored markers, pencils, and regular printing paper for their designs.

We identified several design dimensions and their variations from the different sketches that were produced: the layout of the calendar (linear or cyclic), the positioning of the days and times of the day (on the left or at the top), the shape of drug entries (rectangular, cylindrical, or circular), and the orientation of the calendar (vertical or horizontal). Various sizes, colors, and shapes were also used in the designs. Connecting lines were predominantly used to denote the presence (and absence) of conflicts.

We created three designs ($Design_A$, $Design_B$, $Design_C$) by considering i) variations according to these design dimensions, ii) the constraint of compatibility with already existing calendars, and iii) the intention to remain as close as possible to the design of regular medication schedules. The three designs cover a range of design variations regarding layout, representation of medication entries, and representation of conflicts. This allowed us to assess the usefulness of design variations at a component level, rather than at an overall design level. The following subsections discuss the resulting three design variations according to our usability requirements.

### 4.2  Calendar Layout ($R_{layout}$)

We employed the linear layout [44] for all designs. We discarded the radial layout to stay as close as possible to both regular calendars [53, 54] and medication schedules [66]. $Design_A$ and $Design_B$ show the days of the week at the top of the calendar and the time of day on the left, like most digital calendars. $Design_C$ shows the days of the week on the left and time of day at the top, like most medication schedules. All designs showed the day of the week (in abbreviated form), the month of the year, and the year.

To differentiate headers from entry cells, headers have a grey background whereas cells have a white background. Each design highlights the current day: $Design_A$ does so with a bold and bigger font size like in MS Outlook [53]), $Design_B$ with a red font color, and $Design_C$ with a grey background like in Google Calendar [54].

### 4.3  Calendar Entries ($R_{medic}$)

All three designs show entries using rectangular shapes like most digital calendars do. However, design variations in terms of position, color and size were explored with each calendar design.

$Design_A$, shown in Figure 2, maintains the layout of existing calendars such as Google Calendar [54] and MS Outlook Calendar [53]. The height of rectangular entries indicates their duration, color is used to differentiate types or categories of entries (as set by the user), and their names are conveyed with textual labels. In this design, we also represent medication entries with rectangles (or bars), whose vertical position and height indicates start and end of the allowed administration period for the medication. Medication entries have an embossed horizontal marker placed at some point along the bar to indicate the planned administration time (at which point the reminder would trigger if programmed). Preferred administration time of a medication entry is shown with higher opacity and allowed administrative time with lower opacity. Color hue encodes the type of medication.

$Design_A$ supports medication (or drug) entries and physical activities. Each drug entry in the calendar is labelled with the name of the drug and suffixed with bracketed drug dosage. The suffix -WF indicates that the drug should be administered with food. Physical activity entries have a full-color fill, a dashed border, and a label indicating the name of the activity. All other calendar entries are represented with rectangles filled with different shades of grey.

$Design_B$, shown in Figure 3, maintains the layout of existing calendars and maintains the standard way of representing calendar entries. As such, this design consists of not altering the representation of the existing base calendar and its entries, and to *overlay* a representation of medication entries. $Design_B$ has the same layout as $Design_A$. Medication entries, like other entries, are shown with rectangular shapes. They have no fill color, have an outline color hue that conveys the type of medication, a solid outline or a dashed outline if they represent drug or physical activity respectively, and a label that displays the name and dosage of the medication. A filled circle and protruding vertical bar (to illustrate a spoon) in the top-left corner of the rectangle indicates whether the medication should be taken with food or not. A small filled rectangular inset indicates the period of the day when the medication should be administered through its vertical position and height. No marker indicates that the medication should to be taken during the hour on which the entry is positioned, a full-height marker indicates that it can be administered at any time of the day, and a partial-size marker specifies the period of the day that it should be administered (top for morning, middle for afternoon, and bottom for evening). This encoding makes it possible to visually convey larger time periods for administration without cluttering the calendar with large rectangles.

$Design_C$, shown in Figure 4, has a layout different from $Design_A$ and $Design_B$ as it maintains the layout of a common medication schedules. It shows days of the week on the left and times of the day

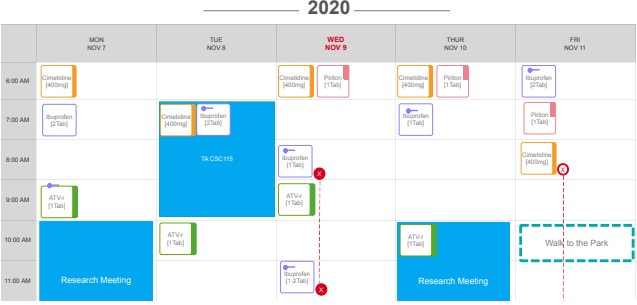

Figure 3: $Design_B$ with marked rectangles bearing a colored outline used to represent medication entries. Rectangular entries with dashed outlines represent Physical Activities. Blue full-width rectangular entries show other activities.

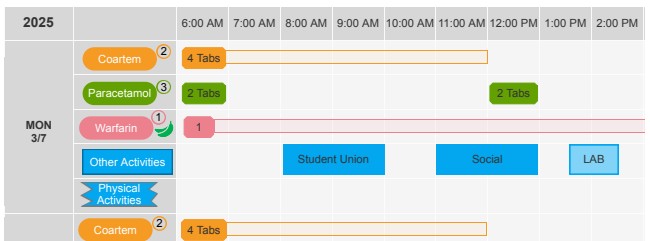

Figure 4: $Design_C$ with medication entries represented as sliders that communicate *allowed* administration time interval and preferred administration time. Each medication entry has its own row. Rectangular entries with and without rugged side-edges represent physical activities and other activities, respectively.

in columns at the top, giving a layout with daily entries that read from left to right (as opposed to top to bottom). Each row in the calendar corresponds to a medication entry within a day.

The first column of a row shows the name of the medication, the number of times the medication should be administered that day, and possibly an indication if the medication should be administered with food (visualized with a banana icon). The medication dosage and administration time are represented with a slider. The position and length of the slider indicate the period in which that medication can be taken. The rectangular buckle indicates the preferred time.

Two additional rows are displayed: one for physical activities, and one for all other, standard calendar entries. They are visually differentiated by using different types of edges. The start and width of a block indicates the start and duration of the activity.

### 4.4 Conflict Visualization ($R_{conflict}$)

Visualization of conflicts between medication entries is shown differently in the three designs, considering i) that conflicts can be of two types (too-apart or too-close) and ii) the results from the ideation session (color and connecting lines were dominantly used to denote conflicts). Figure 5 shows how conflicts are shown in each design.

Some designs featured rectangular enclosures for conflicting pairs. For B and C, we used lines to show conflicts and match conflicting entries because lines do not overwhelm the calendar (and would generally be effective in connecting points as employed in graphs [73]). For A, we matched the conflicts using numbers.

$Design_A$ stays as close to the features already supported in the calendars as possible. As such, we used fill and stroke to denote the presence of conflicts between two entries. An entry that is flagged as being part of a conflict has its block filled with red (for too-close) or its outline changed to red (for too-far-apart). Numeric labels indicate the conflict identifier i.e., two entries marked with the same number

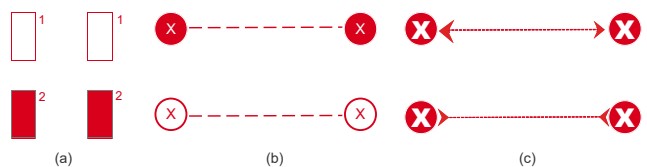

Figure 5: Conflict representations: (a) filled and outlined entries for $Design_A$, (b) dashed lines with filled and outlined circles for $Design_B$, and (c) directed solid and dotted lines for $Design_C$.

are part of the same conflict.

$Design_B$ indicates conflicts between entries with connecting red dashed lines – an approach that is effective in showing connection between points as employed in node-link diagrams [73]. Start and end of line markers convey conflict type: a filled circle with a cross mark inset indicates a too-close conflict, and an outlined circle with a cross mark inset indicates a too-far-apart conflict.

$Design_C$ indicates conflicts using connecting lines like in $Design_B$. However, in this design the line style encodes the type of conflict (solid line with arrows pointing outwards for too-close and dotted line with arrows pointing inwards for too-far-apart). Start and end of line markers are filled circles with a cross mark inside.

## 5 STUDY RATIONALE AND METHODOLOGY

We designed three prototype calendars with the aim to study design variations of layout, representation of entries, and representation of conflict. The goal of this study is to assess the quality of design variations, rather than the quality of each design as a whole.

We hosted the three designs on a public server and conducted remote moderated think-aloud sessions via Zoom. Participants were asked to complete a series of tasks with each design variation. The order in which the designs were presented to the participants was counterbalanced and each participant was randomly assigned an order. Each session lasted for approximately one hour. The study was approved by our institutional ethics board.

### 5.1 Participants

Our recruitment was informed by the focus of this research on active people with frequently changing appointment schedules and who manage medication prescriptions [74]. The recruitment was targeted at people who were between the ages of 35 and 65, who were taking (or helping someone taking) medication regularly, and who considered themselves to have a reasonably busy schedule. We recruited participants through social media platforms, university mailing lists, and posters displayed on campus. Those who were interested reached out to the researcher and were recruited on a first-come-first-serve basis. Participants were compensated with a CAD20 Amazon e-gift card. Twelve participants (P) were recruited and completed the study (10 female, 2 male). They were aged between 35 and 54 years old. All participants were either taking and/or helping someone taking multiple medications: nine were on medication, two were both on medication and helping another person on medication, and one was only helping another person with their medication. There were five students, two nurses, three information and communication technology workers, one human resource consultant, and one professor. Although participants self-selected based on our criteria (we did not question their perceived business), the frequency at which they used a calendar was a proxy for us to measure this. The frequency with which participants used a calendar to manage their schedule was daily (9), very often (2) and none (1 – this participant indicated frequent use of Medication Administration Record).

Table 1: Tasks participants performed with each of the designs, grouped by usability requirement each task addresses. The last two columns indicate which measures were collected for each task.

| Usability Requirement | Task Identifier | Task Description | Correctness | Completion Time |
|---|---|---|---|---|
| $R_{layout}$ | $T_{LAYOUT;year}$ | Reading the current year | x | x |
| | $T_{LAYOUT;month}$ | Reading the current month | x | x |
| | $T_{LAYOUT;week}$ | Locating the day of the week | x | x |
| | $T_{LAYOUT;day}$ | Identifying the current day | x | x |
| | $T_{LAYOUT;hour}$ | Locating the hour of the day | x | x |
| $R_{medic}$ | $T_{MED;entry}$ | Identifying medication and non-medication entries present in the calendar | x | |
| | $T_{MED;day}$ | Reading medication to be administered on a given day | x | x |
| | $T_{MED;repeat}$ | Reading the number of times a medication is repeated in a given day | x | x |
| | $T_{MED;cycle}$ | Reading other days of the week when a given medication was administered | x | x |
| | $T_{MED;dosage}$ | Reading the dosage of a given medication | x | x |
| | $T_{MED;food}$ | Indicating whether a given medication is to be taken with food or not | x | |
| | $T_{MED;slots}$ | Identifying alternative slots for a given medication entry | x | |
| $R_{conflict}$ | $T_{CON;entry}$ | Identifying conflicts that are present in the calendar | x | |
| | $T_{CON;med}$ | Naming the medication involved in a conflict | x | |
| | $T_{CON;type}$ | Determining the type of a conflict | x | |
| | $T_{CON;resolve}$ | Suggesting actions for resolving an identified conflict | x | |

## 5.2 Study Procedure

The study began with the participants agreeing to the consent form. Then, they were asked to provide demographic information and were provided a URL to access the designs from their browser. They went through a familiarization phase during which they could study the guide (legend), the layout, and the content of the three designs. The participants were encouraged to think aloud throughout the session.

The participants were then asked to complete a series of tasks with each design. The participant performed 16 tasks: 5 addressing $R_{layout}$, 7 addressing $R_{medic}$, and 4 addressing $R_{conflict}$. Table 1 summarizes these tasks. After they had completed each task, the participants were asked whether there were features of the design that they found either useful or not useful to complete that task. After tasks $T_{MED;entry}$ and $T_{MED;day}$ under $R_{medic}$, and tasks $T_{CON;entry}$ and $T_{CON;type}$ under $R_{conflict}$, participants were asked to evaluate the difficulty of performing the task. We selected these tasks because other tasks within each usability requirement are dependent on them. Once a participant had completed all tasks for a given design, they were asked for design suggestions before moving to the next design.

At the end of the session, participants were asked which design they preferred overall and how likely they were to adopt any of the designs for the management of their medications. They were given an opportunity to suggest improvements to each design in order to better suit their medication management routines and asked if they had any concluding remarks before the session ended.

## 5.3 Data Collection

We recorded audio and video streams of the sessions.

The audio component also included three verbally administered questionnaires.

The *demographics questionnaire* was completed at the beginning of the session. The *difficulty questionnaire* asked participants to indicate the difficulty of: (i) differentiating medication entries from non-medication entries ($T_{MED;entry}$), (ii) reading the medication to be taken at a given time ($T_{MED;day}$), (iii) identifying the presence of conflicts in the schedule ($T_{CON;entry}$), and (iv) knowing the nature of the conflict ($T_{CON;type}$). Answers to these questions were provided on a 7-point Likert scale (Very Easy, Easy, Somehow Easy, Unsure, Somehow Difficult, Difficult, and Very Difficult). The *adoption questionnaire* was for evaluating the likelihood of using a calendar that employs visualizations to show schedules that include visualization of medication prescriptions alongside other activities. Answers

to these questions were provided on the same 7-point Likert scale.

We recorded the correctness (measure of success), and the completion time for tasks with succinct answers. We do not report completion time for tasks that had participants provide descriptive answers that could range from one or two seconds to dozens of seconds, as this would report the time it takes to provide a description and not the time it takes to find the answer to a question.

## 5.4 Data Analysis

We performed a qualitative analysis of participants' think-aloud, reactions, opinions, and suggestions. We transcribed the entire sessions using Otter [75] and used NVivo [76] for the analysis. We employed three rounds of coding: first open coding (identifying any interesting concepts), then selective coding (grouping the concepts), and finally axial coding (relating the concepts) [77].

In the open coding stage, one researcher analyzed the transcribed data and coded the data according to the task addressed. The codes from this round were grouped according to the design referenced.

The second round of coding involved carefully examining the text to identify emerging concepts across designs. A single participant's data was selectively coded by the same researcher to gain insights into the depth of user sentiments on various aspects of the designs. A second researcher who was not involved in the study independently coded the data for the same participant. Notes were then compared and the categorization and naming conventions used for the codes were harmonized. This stage resulted in 34 codes.

These codes were further analyzed in a round of axial coding to identify and define relationships between them. This analysis resulted into three categories: *temporal features of the calendar*, *design of medication entries*, and *rendering of conflicts*. These findings are discussed in detail in the results section (below). We structure the report of the study results under these three themes.

To complement the qualitative analysis, we analyzed the quantitative data (correctness and completion time) using descriptive statistics. We used this approach because of the small sample size. The small amount of data collected yields too low statistical power to confidently draw conclusions between design variations. We conducted the quantitative analysis using Tableau Software [78] and relied on the median as the measure of central tendency.

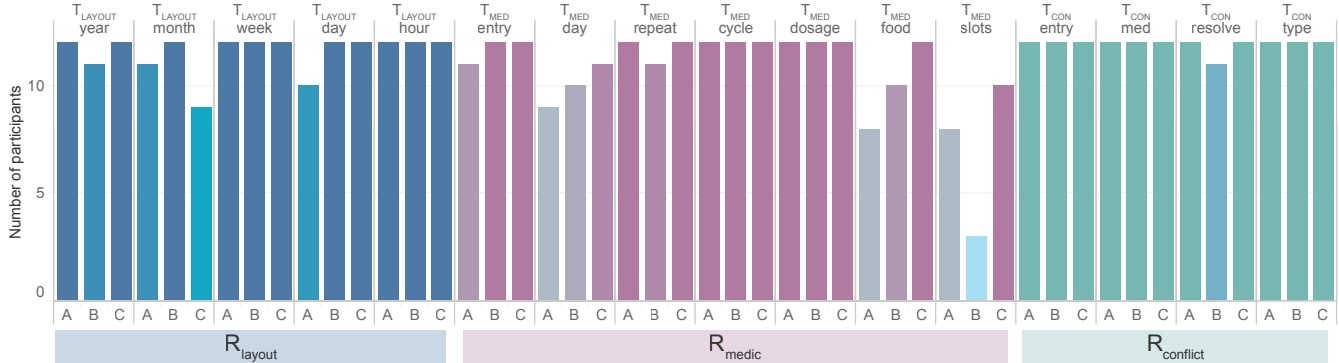

Figure 6: A summary of task completion for the three designs A, B, and C. Tasks in $R_{conflict}$ had the most successful completion rate. $T_{MED;slots}$ was the most failed task with Design B recording the lowest score from the entire study.

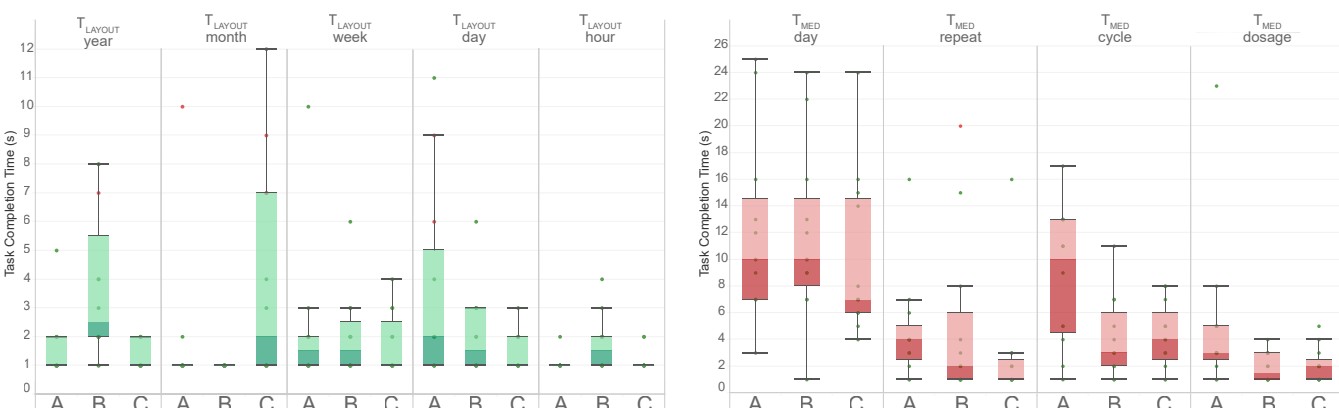

Figure 7: Box-plots showing task completion times for five layout tasks ($R_{layout}$- Left) and four medication tasks ($R_{medic}$- right) for the three designs. Each box-plot shows the minimum time (lower whisker), first quartile (Q1), median, third Quartile (Q3), and maximum time (Upper Whisker).

## 6  STUDY RESULTS

We first report the study results for each task using completion rate and completion time as well as contextualizing qualitative feedback. Then, we present the participants' preferences regarding the different designs and individual design elements.

### 6.1  Task-based Results

We break down the task-based results according to our three task categories: those that deal with the temporal features of the calendars and address $R_{layout}$; those that deal with the design of medication entries and address $R_{medic}$; and those that deal with the rendering and conflicts and address $R_{conflict}$. Figure 6 shows task completion (correctness) for all tasks and for each design. We report on completion times for all tasks addressing $R_{layout}$ and for four of the seven tasks addressing $R_{medic}$ (see Figure 7). Figure 8 shows participant's evaluation of the difficulty of performing tasks $T_{MED;entry}$, $T_{MED;day}$, $T_{CON;entry}$, and $T_{CON;type}$.

#### 6.1.1  Reading Temporal Features

Here we present the results for the five tasks that addressed $R_{layout}$. Correctness and completion time were recorded for all five tasks.

Most participants successfully completed all $R_{layout}$ tasks quickly with all designs. No comments were made about $T_{LAYOUT;week}$, $T_{LAYOUT;hour}$, and $T_{LAYOUT;year}$.

For $T_{LAYOUT;month}$, five participants (P3, P4, P6, P8, and P9) commented that showing the month with a number in $Design_C$ made reading the date difficult because they could not tell which number represented the month and which one represented the day. For example, P6 said *"I don't like the fact that March is represented by*

a number. I find that confusing when I've got two numbers slashed beside each other."*.

For $T_{LAYOUT;day}$, two participants (P3 and P6) thought the larger font size used to indicate the current date in $Design_A$ was an error. For example, P6 said *"That's why the 21 is so big. I was gonna say there's an error. There's something weird going on with the 21. Maybe Friday was a special day and it gets to be big."* P3 proposed that instead of just showing the current day, there should also be an indication of the current hour, saying *"So right now we're in the 10 o'clock block. So to have maybe a dotted outline around that block, just to let the user know, this is the time of day."* P3 also suggested that as opposed to an indication of the current day on the header, the entire column should be highlighted, and that the header should be floating to remain in place when scrolling.

#### 6.1.2  Reading Medication Entries

Here we present the results for the seven tasks that addressed $R_{medic}$. Correctness was recorded for all seven tasks, and completion time was recorded for those with non-descriptive answers ($T_{MED;day}$, $T_{MED;repeat}$, $T_{MED;cycle}$, and $T_{MED;dosage}$).

Most participants successfully completed the $R_{medic}$ tasks with all designs. These tasks were longer to perform than the $R_{layout}$ tasks.

All participants successfully completed $T_{MED;entry}$ with all designs. Once they had provided their answer, participants were asked to describe which visual variables ( color, size, position, and shape) they relied on to provide their answer. Participants relied on color only (6 with $Design_A$, 2 with $Design_B$, 3 with $Design_C$), shape only (3 with $Design_A$, 5 with $Design_B$, 7 with $Design_C$), size and shape (1 with $Design_A$), shape and color (1 with $Design_A$, 4 with $Design_B$),

and shape and position (1 with $Design_C$). One participant did not rely on any visual variable to provide their answers with all three designs. P6, P7 and P4 used the name of the medication to identify medication entries in $Design_A$, $Design_B$, and $Design_C$ respectively. For example, P7 said after they had completed the task with $Design_B$, *"I can tell which ones the medication entries are by the names of the medications. I also noted that there was a dosage size or a number of tablets."*. One participant (P9) indicated that the difference between medication and non-medication entries was not clear until they read the legend, saying *"For the first moment it was not so clear. But after [reading the legend] the shape is thinner than the [one for] regular meetings."*

All participants successfully completed $T_{MED;day}$ with all designs in similar times. Three participants (P3, P5, and P7) complained about the need to scroll to the end of the day with $Design_A$. For example, P3 said *"I find it's a lot of scrolling down. It would be helpful if there was a way to condense it or to make it possible to see the entire calendar available in terms of morning, afternoon, and evening."*, and P8 said *"The time frames are a bit big. So it makes like I said, it really makes it scroll off that you can't see it all in one consolidated view"*. With $Design_B$, participants were expected to use the daily medication summaries provided at the top. Three participants (P1, P6, and P9) found the daily summaries helpful in performing this task. For example, P9 said *"Yeah, I like the idea of having the first row on the calendar dedicated only for the medications that needs to be taken. I think it brings an overall idea [of] what should be taken during that day."*. Three participants (P3, P4, and P5) also complained about the lines demarcating days not being clear. For example, P4 said *"I have a harder time differentiating the calendar component the days, because there's not a strong border between the days of the week."*.

All participants successfully completed $T_{MED;repeat}$ quickly with all designs. Similar complaints about the need to scroll as for $T_{MED;day}$ were made about $Design_A$ by P3, P5, and P8. With $Design_B$, participants could count the number of dots beside the name of the medication in the daily summaries at the top. Five participants (P3, P6, P7, P9, and P10) complained that the dots were not intuitive. For example, P6 said *"So why is there three circles for purple and one circle for green and one circle for Orange? I'm not sure what that means. It's not immediately intuitive"* P3, P7, and P9 thought that the dots indicated the maximum number of pills that was supposed to be taken per day, not the actual number of pills that should be taken. For example, P7 said *"And then these numbers in the circles next to the names of the medicines, I don't know if that means the maximum number of times allowed per day. So the numbers don't really make sense to me."* No participant had trouble with $Design_C$, where the number was directly indicated beside the name of the medication.

All participants successfully completed $T_{MED;cycle}$ with all designs. With $Design_A$, they again had to scroll through the entire week, therefore took longer; with $Design_B$ they could rely on the daily summaries; and with $Design_C$ the information was readily accessible on the row headers. No comments were made regarding this task.

All participants successfully completed $T_{MED;dosage}$ quickly with all designs. Four participants (P1, P4, P7, and P10) commented on the inconsistency in the unit used for the dosage, saying that only milligrams (mg) should be used. For example, P10 said *"It's a little bit inconsistent that it's milligrams of Metformin, but not for the Advil [...] those could be in milligrams too."*.

Most participants successfully completed most $T_{MED;dosage}$ with all designs. 10 participants commented that an indicator for take-with-food medication should use a food icon (as used in $Design_C$). For example, with $Design_B$ P3 said *"I'd love to see a symbol that is food-related, as opposed to a slider-style tab."*. P7 indicated that the $WF$ (for "With Food") suffix used in $Design_A$ could also be read

as "Without Food". P7 also thought that when a food icon is used (such as in $Design_C$), it should be a realistic icon and alternatively be personalizable, saying *"But I would want to make it look more like a banana. If that's supposed to be banana, they're green, but bananas are typically yellow. [...] It'd be fun if someone could choose the emoji they want to use or the picture they want to use to indicate this action"*.

Participants had difficulty completing $T_{MED;slots}$, especially with $Design_B$. To complete this task, participants could rely on the bars that indicate allowed medication intake times with $Design_A$ and $Design_C$. With $Design_B$, the marker on the medication entry indicated the allowed time and five participants said $Design_B$ did not support that task. For example, P1 said *"it doesn't show any other time of the day that you can take it"*. Three participants (P5, P11, and P12) said they could reschedule in any free slot, for example, P9 said *"It seems that 7am is a possibility because there is no other and there is no indication of conflicts."*. P10 commented they would move it to a slot and observe if a conflict was flagged, saying *"I don't know. I think I would just move it and see if a conflict came up."*. P8 indicated the use of the bar to indicate allowed schedule times could also be read as extended release time, saying *"That would mean that it's something that it's an extended release. Warfarin is not an extended release."* P10 and P6 reasoned that a medication that is supposed to be taken at a specific time point should not occupy a full hour on the calendar. For example, P10 said *"I really don't like this fact that it says 6am on the side and then it makes it a block of time."*

### 6.1.3 Reading Conflict Annotations

Here we present the results for the four tasks that addressed $R_{conflict}$. Correctness was recorded for all four tasks, but not completion time because they all asked for descriptive answers. Overall, participants had no difficulty completing these tasks and they did so quickly.

Yet, participants provided comments about the designs in light of completing these tasks. For example, P4 said *"At first glance, it did not seem clear. It looked unfriendly just because there's red and axes and stuff. But talking through it with you, and I was trying to explain the differences, it made more sense."* For P8, the fill and outline used for $Design_A$ was not as clear at with the other two designs. They said *"I think switching the coloring doesn't necessarily work to bring out the conflict because you lose the fact that you expect to see this green with blue border for Advil, and now I'm instead seeing this red with green border. Is that a different pill?"* Two participants (P3 and P6) thought the design should not allow users to schedule in slots that would cause conflict, for example P6 said *"You're supposed to take one tablet every four to six hours. They shouldn't have scheduled it there."*. Four participants (P1, P10, P3, and P8) said the designs should show available slots and flag an error when the user tries to schedule a medication entry in an illegal slot. For example, P10 said *"So I don't know if there wouldn't be like, the potential spots, if you're going to ask them to move it like where you could have included the constraints to make them blank spots in there."* The four participants (P3, P8, P7, and P10) who noticed that information about the conflict was available when hovering over the conflicting entries with $Design_A$ and $Design_C$ were positive about the feature. For example, P7 said *"when you hover over it, it gives you more information about the conflict that you're having. I do like that."*

## 6.2 Design Preferences

When asked about the likelihood of using a calendar that integrates medication entries, eight answered positively (1 somehow likely, 3 likely, 4 very likely) and four answered negatively (1 somehow unlikely, 1 unlikely, 2 very unlikely). When asked which design they preferred, 1 answered $Design_A$, 10 answered $Design_B$, and 1 answered $Design_C$. Below we present participants' rationale for their reservations and preferences.

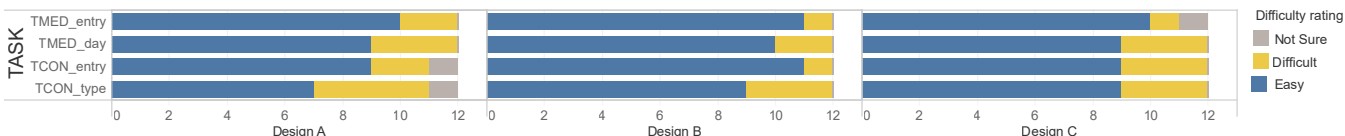

Figure 8: Participant's evaluation of the difficulty of performing tasks $T_{MED;entry}$, $T_{MED;day}$, $T_{CON;entry}$, and $T_{CON;type}$ for the three designs.

Four participants had **reservations** to adopt such a calendar. They provided different reasons for this. P6 wanted to avoid having a constant reminder of medication in their calendar. They said *"I live by calendars. But I have so much going on in my calendar as it is, if I start putting my medication in there as well. It gets way too much emphasis."* P7 was satisfied with their current paper-based system. They said *"Because I have a system that I enjoy using on my own. I have my pills in a pill case. I have a list that I can pull up at any time, that will remind me how many times per day what the dosages per day."* P8 did not see the need of a calendar for medication, because in their case all their medications are to be taken at the same time daily. They said *"I have my medications [which] I take once a day. I make sure that myself, it's the first thing I do when I get up in the morning. It's one of the first activities."*

**Familiarity** (or unfamiliarity) was one reason why participants opted for one design as opposed to another. This was the case for five participants (P7, P6, P5, P4, and P10). For example P7 commented about $Design_C$, *"I haven't actually used a calendar that looked like this in quite some time. And so right now, for me, this isn't a design that's useful"*. P5 said about $Design_B$, *"I think this is more of a calendar view that I'm used to seeing items in slots like this. So that's familiar."*, and P4 said about $Design_B$ *"This one reminds me a little bit more of a Google Calendar, which I'm familiar with."*

**Clutter**, or lack thereof, was another reason for favoring a design raised by 6 participants (P1, P2, P5, P6, P8, and P9) indicated that $Design_A$ and $Design_C$ were too cluttered. For example, P8 said about $Design_C$: *"My first thing seeing it, it's very cluttered. It's got a lot of information on it."* P6 thought that medication entries were given too much space at the expense of other entries, saying *"Well, I don't like again, but there's so much emphasis on the medication, it's almost like the medication takes precedence. And everything else that you've got going on your life is almost insignificant."* P1 suggested toning down medication entries, saying *"instead of having the name of the medication, just put four tablets."* P2 found that too much color was used in $Design_C$, saying *"I think there are a lot of colors here, which makes me confused."* P6 found that the squares used for $Design_A$ could be made smaller, saying *"We [should] make the squares a bit smaller somehow or less predominant, less colorful."*

One (P10) participant raised **privacy** concerns. They suggested that the entries should be discrete enough to avoid exposing information in the event that someone else accessed the calendar. *"So I don't know if it would be something that was like, more discrete or if someone was to accidentally see my calendar and like, saw that I was taking all these medications or something like that."*

### 6.3 Other considerations

Participants described a few features they thought were missing in the designs. These included the timing of reminder, the inclusion of a notation for medications that should be "taken as needed", and the inclusion of "over the counter" medication. P8 said that there should be feedback mechanism embedded into the system so that the calendar should confirm that the user has taken the medication. They asked: *"I guess to all this calendar stuff is when you need to take it. And then did you actually get confirmation of taking it?"*

## 7 DISCUSSION AND DESIGN GOALS

The results from our study confirm that prescriptions can be incorporated into mainstream calendars to allow for management of medication prescriptions (**DG1**) and that patients would generally be willing to adopt a calendar system that supports this aspect. Our results also show that the design of such a calendar should not deviate too much from the way conventional calendars are designed and that medication entries should be integrated with other non-medication calendar entries. They should not occupy too much space and should be dismissible by users.

Below, we discuss the results under the five following themes that emerged from analyzing the qualitative data: use familiar design, avoid clutter, allow for personalization, support personal reflection, and highlight for attention. Each theme informs a new design goal for integrating prescriptions into calendars.

### 7.1 Use Familiar Design

*The design of a calendar should not radically deviate from calendar interfaces that users are familiar with (**DG2**).* This was observed in various aspects of the design such as layout, medication entries, and icons used to annotate entries such as those which should be taken with food. Over 80% of the participants preferred $Design_B$ because of its probable similarity to already existing calendars. This was surprising to us because $Design_A$ was the design that was intended to resemble existing calendars. The sidelining of $Design_A$ is attributed mainly to the height of medication entries which spanned the entire allowed administration period and introduced too much clutter. The results indicate that the preferred layout should be vertically oriented with days of the week at the top and times of the day on the left.

The dosage used on the medication entry should be one that users are familiar with. The unit used should be consistent with the one used in the prescription. It should show the actual quantity (e.g., milligrams) as opposed to relative classifications such as number of pills or tablets. Similarly, realistic food-related icons (e.g., a banana) should be used to denote that medication that should be taken with food. Such icon should be positioned together with the entry and not as part of medication summaries.

When dealing with conflicts, arrows are effective in communicating the suggested conflict resolution action. End-of-line arrows can be used to indicate that medication entries that have been scheduled too close together should be taken apart and vice-versa. The calendar should also have support for indication that a given entry is optional. Such entries would be used for medications that should be administered "as needed" and non-prescription medications that are sold "over the counter". These design decisions are influenced by everyday activities that users are familiar with.

### 7.2 Avoid Clutter

*The design of the calendar should avoid design elements that introduce clutter to the calendar (**DG3**).* One of the reasons why $Design_B$ was preferred is because it is less cluttered: medication entries can be rendered effectively using position, shape, and size. The size of a medication entry should be as small as possible so as not to occupy too much space. Size should not be used to indicate either allowed or preferred administration time of a medication entry, and the size of the entry should also be uniform regardless of the length of the allowed period of administration. Using shapes with a

colored outline and transparent fill was associated with less noise by participants. While the slider design was effective in communicating both the allowed and preferred administration period, it made the entry occupy a lot of calendar space and was also misread by some participants. Familiar icons such as tablets can be used to indicate medication entries.

Color should be made less dominant and should not be used as the primary identifier for medication entries. While solid fill color was effective in indicating busy slots, using color fill for medication entries was cluttering the designs. The amount of medication information shown (e.g., labels, including name and dosage) should also be minimized. Labels were a source of confusion as to which entry they referred to when multiple entries occupied the same cell. They should be abstracted from the overview and instead be made available as details on demand.

Conflict overlays were easily identifiable on all the designs. Participants preferred the use of indicators for the position of medication entries that are involved in the conflict. The connectors (lines) for conflicting entries should use thin or dotted lines rather than thick solid lines. Participants found that different line styles may appear similar at a distance and hence fail in communicating the nature of a conflict.

### 7.3 Allow for Personalization

To design an effective calendar, we need to tailor the design to individual needs, values, and preferences [79]. Therefore, *the design of the calendar should have provision for users to personalize some of its features (DG4)*. Such desirable personalization includes adding color to medication entries, choosing icons to be used for medications that should be administered with food, deciding which medication information to display on the entry, and choosing whether to use the default entry shape or substitute it for other Emojis or Icons. Personalization should also cover data privacy. The design of the calendar should have features that will protect the users' sensitive medication data from unauthorized reading. This is particularly important when a calendar is accessed by more than one person. Calendar owners should be able to hide features of the calendar that they do not wish anyone without privilege to see.

### 7.4 Support Personal Reflection

*Medication entries should have separate designs for entries that are future and those that are past (DG5)*. Past entries should allow for reflection of past medication-taking behavior by confirming whether the user took the medication or not. The design should therefore have a way of letting users confirm that they have taken the medication to aid personal reflection.

### 7.5 Highlight for Attention

*The calendar should highlight entries to which the users' attention should be drawn on any given day (DG6)*. This includes basic calendar layout requirements: the top of the calendar with the year, month, and weekday labels should be floating so that they are always visible; day separators should be clear and the entire current days should be highlighted; and the current hour should be highlighted.

Medication entries should have markers that communicate the times that their reminders will be triggered. The markers should not communicate time ranges but points in time when reminders are triggered. The calendar should have daily summaries of the list of medications to be administered each day. These summaries should only contain the name of the medication and users should be able to show or hide them.

Medication conflicts should be emphasized on the conflicting entries rather than on the connectors. The user should be notified of a newly created conflict upon rescheduling an entry, preferably via dismissible error messages that describe the conflict. When rescheduling medication entries, cells that are either safe or unsafe should be highlighted to the user to guide their action. Although some participants felt that the design should not allow them to schedule an entry in the space that is likely to cause a conflict, there might be situations where this possibility is unavoidable. The user should, in this case, be guided on possible moves that will resolve the conflict. This can be done by shading or using an outline for all the cells to which an entry may be rescheduled to resolve the conflict, and letting users configure the amount of warnings and error messages they want to receive.

### 7.6 Limitations

One limitation of our study is its relatively small sample size. While 12 participants is appropriate for the qualitative analysis of collected data, more participants are required to make task-based statistical comparisons between designs. That being said, the purpose of this study was exploratory. The findings from this study will allow us to turn to high-fidelity prototyping of calendar designs and to conduct such a quantitative task-based follow-up study.

Another limitation is that since the study was online, we did not have the privilege of observing participant's full activity cycles. It is likely that remote sessions also lead to participants employing less think-aloud than when participating in person. We also constrained participation to people between the age of 35 and 65 who were either on multiple prescription medications or played the role of caregivers to others on multiple medications. While this allowed us to capture insights for that specific population, these insights do not necessarily generalize to other populations. Our calendar designs were suited for relatively large screens such as laptops and tablets and were not evaluated on mobile devices. Given the focus of our study on medication entries, we opted for assigning the same color to all non-medication calendar entries. However, events in real-life calendars are often of several colors. The added colors likely increase visual complexity and visual clutter that must be considered in future studies. Finally, the study was only limited to tasks that relate to reading calendar entries. In the future, tasks such as adding and modifying medication entries should be included.

### 8 CONCLUSION AND RECOMMENDATIONS

In this research, we explored the possibility of integrating prescriptions into calendars (DG1). We considered ways of rendering medication entries so that they were identifiable and that all medication-related information was conveyed. We also considered ways of rendering unsafe medication schedules. We designed three calendars that leverage features available in both conventional calendars and medication schedules. We conducted a study with twelve participants, to evaluate the effectiveness of the calendar designs. Results show that calendars can be designed to support medication prescriptions while remaining familiar to use. Eight out of twelve participants indicated that they were between somehow likely and very likely to use such a calendar to manage their medications. The findings from our study informed five additional design goals that an integrated calendar should address: using of familiar design (DG2), avoiding clutter (DG3), allowing for personalization (DG4), supporting personal reflection (DG5), and highlighting for user attention (DG6). The successful implementation of these design goals will lead not only to a calendar that expressively incorporates all prescription data but to one that is compatible with everyday use of those calendars.

### 9 ACKNOWLEDGEMENTS

We thank the Natural Sciences and Engineering Research Council of Canada (NSERC) for funding this research.

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
