# OpenReview forum: "Integrating Prescriptions into Calendars: Design Considerations"
_graphicsinterface.org/Graphics_Interface/2023/Conference — GI 2023_

### Official Review · Reviewer_aBKb · 2022-12-30
**This paper presents an interesting, exploratory study with 12 participants via Zoom to investigate the design of on-calendar visualization of prescription schedules showing also conflicts arose from medication scheduling. The related work seems relevant and appropriate. The usability metrics used for the study are clearly described. The designs used in the study are problematic and the results should highlight important design implications.**

**Rating:** 6
**Confidence:** 4

**Review:**

However, there is a major flaw in all three designs, each of which uses a single color for “all other activities” while in reality, many people use colors to represent different types of events/activities/reminders in their current calendars, whether digital or analog, e.g., blue for work, pink for personal, green for friends, purple for birthdays and anniversaries, etc. The three designs appear to be different but they are fundamentally the same – using colors to represent different medications and filled/outlines shapes with/without dotted lines for conflicts. Why not consider using the conventional lightning bolt for conflicts/breakdowns? I think the flaw in the color use can be minimized if a user-centered design approach has been adopted from the beginning.

The study results show a need to reduce the use of colors with the designs under study, I wonder how the participants would react to the on-calendar medication schedules if multiple colors are continually used for different types of activities/events/reminders on their existing calendar. Since the study was entirely based on these three designs created by the researchers, the usefulness of the results are unfortunately greatly mitigated. Also, the use of distinctively different shapes may be considered in the redesign.

Moreover, the designs did not take into consideration of whether the medications are new or routine (long-term). This difference should make a big difference in how the medications can be visualized on the calendar. Clearly, detailed information is needed for new meds which are added to existing schedule but routine ones probably do not need such fine details. Another design feature which may be out of the scope of the current paper may be considered is when conflicts are identified, how will they be fixed? If it requires manual interventions, one fix may cause other conflicts, like dominos, which would be overwhelming to the users. What about reminders for medication refills?

The methodology is generally clear but the sample size is very small, which however is not uncommon for exploratory studies. I think the Glaserian Grounded Theory method was used, despite the detailed descriptions of the different coding methods. It’s nevertheless strange that only one researcher performed open coding which is typically conducted by multiple researchers who may then compare the codes for consistency.

The results are clear but mundane with too much details on insignificant findings. I’d suggest streamlining the results section to highlight the more important ones, especially those that are particularly instrumental to the redesign. (X’s in 6.1.2 should be replaced by actual values). Although the design goals identified in the study are not novel, they should be good reminders in the redesign.

---

### Official Review · Reviewer_pivC · 2023-01-09
**Interesting discussion of design considerations**

**Rating:** 7
**Confidence:** 4

**Review:**

There is a lot to like about this submission. The presentation is really well-structured and clear.  The submission provides a good discussion of design considerations and potential solutions.  I liked the balance between keeping the design simple / intuitive while still illustrating the complexity of some of the detailed design decisions.  For example, the submission handles the different ways to depict constraints in an insightful way.  Great job by the authors in ensuring that the participant pool consisted of target population rather than going for a convenience sample. The study tasks appeared to do a good job of getting participants to interact with critical elements of the designs (i.e., provided good coverage of the designs).  Section 6.2 presents interesting qualitative insight; I found these to be the most engaging and important findings from the study.

There are some weaknesses including the following:

- The paper states that the sample size is too small for statistical testing, yet quite a bit of space is devoted to describing the quantitative data and the discussion makes claims about differences in correctness and completion times.  I found this in-between state difficult at times.  One option would be to perform the statistically testing and acknowledge a lack of significant differences (12 participants is small, but it is a within-subjects design).  Another option would be to avoid drawing conclusions based on the numerical differences.  Without the statistical testing, section 6.1 is long for what this data is contributing to the research community.  In addition to some of the conclusions based on the numerical data, the first sentence of the conclusion should be adjusted since the study has not shown that the designs allow for “easy management of mediation prescriptions”

- The paper would benefit from a reconsideration of what the study findings have shown and to have these outcomes better reflected in both the introduction and the discussion / conclusion.  For example sentences like: “Results indicate that calendars can be designed… “ and “conflicts arising from unsafe rescheduling can be rendered” are both too vague and too low of a bar for a research contribution.

- Certain sections of the discussion are long for what they are providing.  For example: “Use Familiar Design” and “Avoid Clutter” are fairly intuitive and widely recognized design goals.  These sections could be streamlined.

Beyond the above, I think there are opportunities to strengthen the paper as follows (some very minor)

- The visualizations in Figure 6 and in Figure 7 make it difficult to visually compare the different designs.  The way colours are used is not helping these visual comparisons.
- Section 6.1.2 has a missing value (i.e., “Median completion time was X”)
- I can see a few factors that might influence the ultimate adoption and utility of a system like this.  One is medication complexity / importance.  For example, some medications have constraints where violating them is not wonderful but is not life-threatening either.  Another factor is the user’s experience with the medications.  For example, someone who has been taking the same medications for years might not want such detailed information on their calendar.
- The privacy issue is interesting and made me think of some of Kirstie Hawkey’s work on incidental information privacy.  I often have to pull my calendar up or have it displayed in situations where others are co-located, where forgetting to “turn off” the medication display would be an unwanted privacy breach.

Overall, I am positive on the work and feel that with some adjustments to wording this paper would be ready for inclusion at GI.  With more extensive editing to the findings and discussion, this could be a very strong GI paper.

---

### Official Review · Reviewer_R8qZ · 2023-01-13
**Great design for visualizing medication (reminders)**

**Rating:** 9
**Confidence:** 4

**Review:**

The paper explored the design of medication calendars for patients to manage their prescriptions.
The paper is well-written and touches on a very important issue. The current apps available for medication tracking often are standalone and do not get integrated into a patient's everyday calendar use. The related work section draws a map of the literature and the gap.
Six main components were listed for administrating medications drawn from the literature.
Three usability requirements were listed to be considered in the design. It is unclear how these requirements were gathered, are they from the literature? Or from interviews with pharmacists?
Based on the identified design requirements, the authors ran design sessions to ideate options. This is an interesting approach. I recommend the authors add more details information about these sessions. For example, how long this session lasted, what materials (pen and paper?) were used? Etc.
The three proposed designs are well-thought. I found the last design, Design C, very innovative and informative. Particularly, it was interesting to see conflict in medications was also displayed in the proposed design. This is an important criterion, especially for patients dealing with multiple conditions.
The evaluation study to test the designs is well executed. One missing piece of information about participant recruitment is on the participant schedule. It was mentioned that “who considered themselves to have a reasonably busy schedule.” I recommend elaborating on how did you measure this to include or exclude participants?
The data gathered from the study is well analyzed. The results about participants’ opinions on the design of the conflicts were interesting and aligned with the literature; often, people find complex vis difficult at first glance, but then slowly learn to work with it and appreciate the value. I enjoyed reading the discussion and interpretation of the results; many design guides came from the study that could help the future design of such systems.  My suggestion is to include related papers to the results that are presented in the discussion. For example, in section 7.3, DG4 talks about personalization. Other work in this area can be cited; some example ref:
[1] F. Rajabiyazdi, et al, "Communicating Patient Health Data: A Wicked Problem," in IEEE Computer Graphics and Applications, vol. 41, no. 6, pp. 179-186, 1 Nov.-Dec. 2021, doi: 10.1109/MCG.2021.3112845.
[2] Walker, Katie, et al. "Visualising Emergency Department Wait Times; Rapid Iterative Testing to Determine Patient Preferences for Displays." medRxiv (2022).
[3] O. Bertelsen, et al. Beyond generalization: Research for the very particular. Interactions, 2018. doi: 10.1145/328942)

Overall, this is a well-written paper that tackles an important issue and offers a series of design solutions to be considered for future work. I argue for accepting this paper.

---

### Meta-Review · Area_Chair_gYFL · 2023-01-14

**Recommendation:** 7
**Confidence:** 4

**Metareview:**

All the reviewers found this paper acceptable for the conference. The three proposed designs are well-thought. (R2) The presentation is really well-structured and clear. The submission provides a good discussion of design considerations and potential solutions. (R3)
Here is a summary of changes that should be made to the paper:
- The source of the three usability requirements used in the study should be described.
- There should be more details about the ideation sessions, e.g., how long each session lasted, what materials (pen and paper?) were used? Etc.
- The inclusion/exclusion criterion for participant recruitment that “who considered themselves to have a reasonably busy schedule” needs to be elaborated on how the busyness was measured.
- There should be justifications for using only one color for all other events and how the findings may be different if multiple colors are used for different kinds of events (as often used, e.g., in Google calendars).
- Would the level of details for routine/long-term and new/short-term medications be different and how?
- Section 6.1 should be shortened given the lack of statistical testing.
- The first sentence of the conclusion should be adjusted since the study has not shown that the designs allow for “easy management of mediation prescriptions”.
- Findings such as “Results indicate that calendars can be designed… “ and “conflicts arising from unsafe rescheduling can be rendered” are both too vague, thus should be elaborated.
- Figure 6 and in Figure 7 should be revised to make them easier for visually comparing the different designs.
- The results and the discussion sections are long with redundant content. Both need to be streamlined while highlighting those instrumental to the redesign.
- The authors should proofread the paper carefully to make sure no missing data.

The following references are recommended to be included in the paper:
 [1] F. Rajabiyazdi, et al, "Communicating Patient Health Data: A Wicked Problem," in IEEE Computer Graphics and Applications, vol. 41, no. 6, pp. 179-186, 1 Nov.-Dec. 2021, doi: 10.1109/MCG.2021.3112845.
[2] Walker, Katie, et al. "Visualising Emergency Department Wait Times; Rapid Iterative Testing to Determine Patient Preferences for Displays." medRxiv (2022).
[3] O. Bertelsen, et al. Beyond generalization: Research for the very particular. Interactions, 2018. doi: 10.1145/328942)